# Challenges of Rabies Serology: Defining Context of Interpretation

**DOI:** 10.3390/v13081516

**Published:** 2021-07-31

**Authors:** Susan M. Moore

**Affiliations:** Department of Diagnostic Medicine/Pathobiology, College of Veterinary Medicine, Kansas State University, Manhattan, KS 66506, USA; smoore@vet.k-state.edu

**Keywords:** rabies antibody, neutralizing antibody, immunity, serological interpretation, assay comparison

## Abstract

The case fatality rate of rabies, nearly 100%, is one of the most unique characteristic of this ancient virus infection. The crucial role rabies virus neutralizing antibody plays in protection is both well established and explanation of why rabies serology is important. Various laboratory methods can and have been used but serum neutralization methods have long been the gold standard due to the ability to measure function (neutralization), however these methods can be difficult to perform for several reasons. Assays such as enzyme linked absorbance assays (ELISA), indirect fluorescence antibody (IFA) and more recently lateral flow methods are in use. Interpretation of results can be problematic, not only between methods but also due to modifications of the same method that can lead to misinterpretations. A common assumption in review of laboratory test results is that different methods for the same component produce comparable results under all conditions or circumstances. Assumptions and misinterpretations provide the potential for detrimental decisions, ranging from regulatory to clinically related, and most importantly what ‘level’ is protective. Review of the common challenges in performance and interpretation of rabies serology and specific examples illuminate critical issues to consider when reviewing and applying results of rabies serological testing.

## 1. Introduction

Survival from rabies virus (RABV) infection is rarely achieved without rabies vaccination, either post- or pre-exposure, largely due to RABV’s exquisite strategies to escape host immunity [1]. Host immune responsiveness to the viral proteins can be achieved through vaccination. Rabies glycoprotein (G) has been well established as the target of effective neutralization by specific antibodies, rabies virus neutralizing antibodies (RVNA) [2,3,4,5,6]. Rabies vaccines provide optimal amounts of rabies glycoprotein to locations, rich in antigen presenting cells (APC) to induce production of sufficient RVNA, as well as memory B and T cells, for prevention of rabies. Vaccination after exposure and administration of passive immunity in non-vaccinated people, in combination with wound cleansing, is highly effective in preventing rabies [7,8]. Current understanding of the RABV pathogenesis and means of prevention and control supports the importance of RVNA in protection of the disease [9,10,11].

RVNA levels are associated with protection, used as proof of efficacy of rabies biologics, and for rabies surveillance. A review of the history of research and development of rabies biologics and the first use of rabies serology, is necessary to explain how the purpose of testing is the starting point in the current applications of rabies serology. The laboratory method used to generate clinical trial immunogenicity data that plays a major role in vaccine evaluations has a direct impact as all methods will not have the same performance characteristics. The mouse neutralization test (MNT) was first used for the evaluation of immunogenicity to rabies vaccination, followed by development of an in vitro assay, the Rapid Fluorescent Focus Inhibition Test (RFFIT) [12,13,14,15]. Early cell culture vaccines were evaluated by analysis of the RVNA response by means of the MNT and RFFIT, both serum neutralization (SN) methods; the RFFIT eventually was determined to be more precise and reliable [16]. Today the RFFIT is the primary assay for evaluation of immunogenicity.

Because RVNA levels are a marker, not the sum of immune protection, establishing a set level of protection is difficult. Innate and adaptive cellular immunity also contribute to rabies protection but methods to measure these immune effectors are less developed and not commonly available, thus RVNA measurement remains the primary means of verifying rabies immunity [17]. Various levels of RVNA have been used for various purposes as a cut-off, but the level 0.5 IU/mL has been most commonly applied. Since 1984, the World Health Organization (WHO) recommends 0.5 IU/mL as the value beneath which a booster vaccination should be given when monitoring individuals who have received pre-exposure prophylaxis (PrEP) due to a continued increased risk of rabies exposure. This recommendation can be traced to a joint symposium of the International Alliance of Biological Standards (IABS) and WHO held in 1977, where participants examined immunogenicity results of several human rabies vaccine clinical trials. The group concluded that a minimum level of 0.5 IU/mL demonstrates seroconversion in a sample collected 4 weeks after completion of the vaccination series [16]. By this moment in history, the 0.5 IU/mL level was found to be associated with protection in dogs and cats (discussed below), however, in humans, this level represents seroconversion (not protection); a point worth noting as clearly there cannot be challenge studies in humans with a virus causing a fatal disease. Yet, this level has been used globally without consideration of test method or time point of sampling, not to mention species, vaccine, and other variables causing rabies serology interpretation to be challenging and, at times, problematic.

## 2. Defining the Purpose of Testing/Context of Interpretation

Rabies and prevention efforts include information gained from scholarly research, pharma industry, sero-surveillance, and vaccine response results from individuals all of which depend on reliable rabies serology results to guide national and community programs as well as inform individual clinical decisions for humans and animals. These efforts and decisions can be sent off track by faulty interpretation of test results. Interpretation challenges can be overwhelming. A review of the history of rabies research and development of rabies biologics is necessary to understand how result interpretation is closely tied to the purpose of testing. Additionally, the purpose of testing is the key factor in method, cut-off and sample selection. Knowledge and appreciation of these three points will aid in interpreting results of current applications of rabies serology. A solid approach is to first understand that the full context of interpretation is achieved by combining purpose of testing with laboratory method performance characteristics. This review attempts to simplify the framing of context needed for full understanding of what the result can tell us, and also what it cannot. The three factors of cut-off level between positive and negative results, the laboratory method used, and the sample chosen for evaluation will be explored concentrating on variables that affect appropriate and accurate interpretation. Figure 1 provides a diagram to display how these three factors relate to each other and to the purpose, as well as the variables of each that provide another tier of consideration.

### 2.1. Cut-Off Level

Immunogenicity testing requires that the targeted cut-off be robustly representative of, as a corollate, the level of protection, not the limit of detection (LOD) [18]. During the advent of modern cell culture rabies vaccines, the RVNA level of 0.5 IU/mL was determined to be a marker of adequate rabies immunization at 4 weeks after completion of the vaccination series, subsequently it has been interpreted to be proof of protection by some. Additionally, 0.5 IU/mL has been used as evidence of specific immune response, or the LOD. While it was not the specific intention of the IABS/WHO group (who made the initial designation of RVNA cut-off in 1977), consideration of specificity and sensitivity were involved in establishing the cut-off. History provides some logic to both interpretations of the 0.5 IU/mL cut-off (as proof of protection and as LOD), though not always appropriately and sometimes incorrectly.

Information available about rabies protection comes from rabies challenge studies in animals. Interestingly, the level of protection determined in dogs and cats by challenge studies is the same 0.5 IU/mL cut-off as determined to be proof of seroconversion in humans [7,19].

The key data instrumental in setting the cut-off comes from two papers. Bunn and Ridpath (1984) presented rabies challenge data in dogs, commenting simply that their “data indicates serology is a valid indicator of rabies immunity”; they did not propose to provide specific recommendations [19]. The RVNA results were reported in the 50% endpoint titer format. The authors concluded an increasing probability of survival was associated with increasing RVNA titer: from 95% at 1:17 to 99.9% at and above 1:127. Following in 1992, Aubert published a summary analysis of challenge studies in dogs and cats (including Bunn and Ridpath’s results converted to IU/mL values), concluding the RVNA levels between 0.1 and 0.2 IU/mL are predictive of survival in dogs and cats [20]. In both the Bunn and Aubert papers, the blood was sampled “pre-challenge.” Similar conclusions were found in wildlife challenge studies; RVNA levels between 0.25 and 0.5 IU/mL were associated with rabies survival, using both the day 28/30 and the pre challenge results, though some species differences were noted [21]. These papers explain the basis of reasoning that 0.5 IU/mL is a protective level. Applying this value without consideration of timing of blood sampling and species differences, not to mention the type of laboratory method renders the use of 0.5 IU/mL liberally across all contexts questionable. Indeed, the OIE recognizes the level of 0.5 IU/mL the minimal value that provides proof of seroconversion after rabies vaccination in dogs, cats and ferrets traveling or exported to rabies-free countries/areas in the world, but also notes “a single cut-off for seropositivity may not be universally applicable among difference species” [22]. Even the breed of a dog has been shown to affect the RVNA response to vaccine, and not necessarily dependent on breed size [23]. What can be concluded from challenge studies is that high circulating RVNA at or shortly after the time of rabies exposure is key to survival, but no definitive level is known to be universally protective. Other immune mediators are stimulated upon RABV antigen exposure. It is the sum of the immune response that determines the outcome of exposure. RVNA is simply a marker of the immune response to rabies antigens and level of immunity long-term. The 0.5 IU/mL was initially intended to assess sero-conversion in rabies vaccinated humans (not protection) and, separately, has been shown to be associated with survival in dogs, cats and wildlife.

Delving further into what factors contributed to settling on the proof of adequate immunization level reveals that the limitations of the MNT and RFFIT test methods, the primary methods to generate the rabies serology results in early rabies vaccine work, played an important role [24,25]. Namely, that non-specific inhibition of the virus was detected in non-vaccinated subjects’ serum at low serum dilutions, a common characteristic of SN assays [26]. In light of this test limitation, and the reality that rabies is a fatal disease, both the WHO and the Advisory Committee on Immunization Practices (ACIP) rabies vaccination guidance groups selected RVNA cut-off levels above detectable levels. Therefore, informally, consideration of the LOD was a factor in setting the cut-off and it should be noted the consideration specifically applied to the assays in use, RFFIT and MNT.

Even as the similar considerations for selecting a level presumed to robustly represent a specific result in SN assays, the WHO and the (historically applied) ACIP recommended RVNA cut-offs are not the same. That the cut-offs are different is not clearly understood. This partly a consequence of the way each guidance expresses rabies serology results [27]. The WHO states the RVNA cut-off in IU/mL values, while ACIP has stated the level either in 50% endpoint (ED_50_) titer format or as “complete neutralization at a 1:5 serum dilution” (neither an IU/mL or ED_50_ titer value). To obtain an IU/mL value of a test sample, the reciprocal ED_50_ titer is converted by comparison to a standard rabies immunoglobulin (SRIG) ED_50_ titer of known potency expressed in IU/mL value. For example, if the Sample titer is 1:50 and the SRIG titer is 1:200 and the potency of the SRIG is 2.0 IU/mL, the calculation is:50/200 × 2.0 IU/mL = 0.5 IU/mL(1)

This standardization of test result is a mechanism to account for the inherent variability of SN assays, resulting in increased precision and accuracy among and within laboratories performing rabies serology globally. Without knowledge of the assay, particularly the ED_50_ titer calculation method, one cannot know that “complete neutralization of a 1:5 serum dilution” is *an ED_50_ endpoint titer value of 1:11*, and further, that *the IU/mL value can range from 0.1–0.3 IU/mL in different laboratories as well as within the same laboratory*. Publications of rabies vaccine response studies have applied levels other than 0.5 IU/mL to analysis of the RVNA levels, potentially causing unequal comparisons of immunogenicity results, which is not always clearly understood by those reading the articles [28,29,30]. It is not a gross aberration for unvaccinated subjects in rabies vaccine clinical trials or sero-surveys to have RVNA levels of 0.1 to 0.4 IU/mL [31,32]. A review of unpublished rabies serology data identified an increasing number of false positive RFFIT results associated with use of decreasing cut-off levels, with a 0.1% false positive rate when 0.4 IU/mL was used as the cut-off to 2.7% using a cut-off of 0.1 IU/mL. The status of rabies as the disease with the highest case fatality rate of any infectious disease uniquely demands that the LOD for immunogenicity determination be securely robust; reporting a false positive can have fatal consequences. Because the 0.5 IU/mL cut-off decreases the risk of reporting false positive RVNA results (improves specificity) while increases precision, accuracy and standardization, it robustly meets the purpose of verifying adequate response to rabies vaccination.

Rabies serology is used for many purposes, and the appropriate cut-off is what best meets that purpose. For many purposes it is not individual protection that is the focus, but concerns such as monitoring for vaccine coverage in a population, sero-surveys of wildlife, or potency measurement of biologics. For use of rabies serology for these purposes, the applicable cut-off is the LOD, defined as the level representing a specific response. The acceptable percentage of false positive and false negative results can vary per the purpose of testing. As described above regarding the WHO and ACIP cut-offs, the 0.5 IU/mL cut-off was not intended to be used as the LOD for all purposes for which rabies serology is used. LOD is determined formally for a laboratory method through validation and may vary by specimen type [33,34]. The cut-off level of 0.5 IU/mL was established based on SN methods for a specific purpose (define adequate response to rabies vaccination at a specific time point) with consideration of the non-specific reactions inherent in SN test methods. Extrapolating this level to other test methods is not supported and can lead to misinterpretation of rabies serology results. Clearly understating what the cut-off represents (and does not represent) requires knowledge of how it was selected and for which laboratory method it was applied.

### 2.2. Laboratory Method

RFFIT and the Fluorescent Antibody Virus Neutralization (FAVN) test are the primary WHO endorsed methods, though other methods, including modifications of the RFFIT and FAVN are in use [7]. Modifying a method has the potential for modifying the performance characteristics. Outside the SN methods of RFFIT and FAVN, various other techniques, such as enzyme-linked immunosorbent assays (ELISA), are used but the same caveat of ensuring appropriate qualification for the purpose applies. Differences in rabies serology methods, performance characteristics, purpose of testing, etc. have been covered in previous publications, notably the difference measuring antibody function versus antibody binding [21,35,36,37,38,39,40]. SN methods have the advantage of providing an in vitro measurement of what occurs in vivo [41,42,43]. That advantage can also provide misinterpretation problems when a method’s linearity over the broad range of RVNA response is assumed. In reality antibody neutralization function is not always linear in laboratory methods leading to an over estimation of RVNA if the method has not been validated for dilutional accuracy [18,44]. ELISA methods include indirect, competitive, or blocking method applications for measurement of antibodies, each with their own unique capabilities and modification/optimization choices just as with SN methods, consequently interpretation will be dependent on these capabilities. For example, a paper comparing the immunogenicity of rabies vaccine regimens using RFFIT versus an indirect ELISA demonstrated that equivalence determination can vary depending based strictly on which assay is used to measure immunogenicity, primarily due to the difference of immunoglobulin classes detected by each method [45]. Likewise, determination of bio-equivalence of two rabies vaccines made with non-identical parent strains can differ based solely on use of a homologous or heterologous challenge strain in the test method [46]. One can only imagine further risks of misinterpretation if variables of the test method type and performance are not understood and controlled. An increase in street rabies virus variants with decreased identity to vaccine parent strains since 2010 globally, including mutants resistant to neutralization with vaccine induced RVNA has been reported [47]. Reduced homology between the vaccine and the exposure strain potentially impacts the antigen source chosen for the test method selected to demonstrate an adequate immune response for those at risk of encountering those strains. The advent of monoclonals to replace human and equine polyclonal rabies immunoglobulin (RIG) products and new vaccine types, such as mRNA or vector vaccines, adds complexity to the consideration of method selection, particularly regarding specificity capabilities (breadth of coverage). For the passive immunity component of post exposure prophylaxis, a monoclonal (or monoclonal cocktail) must be able to neutralize all RABV strains the target population may encounter. The antibodies induced by a single version of the RABV glycoprotein, the target of RVNA, similarly must be able to neutralize the range of antigenic variation of any potential exposure. Specificity and the LOD are two major factors in interpretation depending on the purpose of testing.

Because the LOD is the lowest level detected attributed specifically to the analyte of interest, any factor causing a non-specific signal will affect this level [26,33]. Causes of non-specific reactions and interference can be unique to the particular type of assay; an insufficient washing step in ELISA or poor cell health in SN assays, or more generally the presence of cross reacting antibodies in the specimen [26]. Cross-reactive antibodies can be found within a virus family as was illustrated for flavivirus in a recent paper [48]. Though cross-reactive antibodies have been shown to be not long lasting, they can still confound serology results as was found in cross-reacting Dengue and Zika antibodies [49]. With more species of Rhabdoviruses being discovered with molecular methods, such as next-generation sequencing, it is possible that antibodies specific to non-pathogenic Rhabdoviruses may be present in serum samples, demonstrated recently in human serum in Africa for the Tibrovirus species Ekpoma-1 and -2 viruses, which may be a source of cross-reacting antibodies [50]. Even outside virus families, if the homology between the target protein is high enough and the confounding antibody level is high, cross-reactivity is possible as in the case of a false positive HIV antibody result in a rabies vaccinated blood donor with a very high RVNA level [51]. Additionally, a study revealed interfering antibodies to other infectious viruses in the serum of acutely ill patients as the cause of false positive results in rabies serology tests [52].

Without defining the performance characteristics of a method and specific modifications, including most importantly the cut-off level relevant for the purpose of testing (whether a level that reflects/predicts protection or specific detectable antibody response) one cannot have all the information needed to make reasonable interpretations of the results. Besides the method and the purpose, the sample itself is a consideration.

### 2.3. Sample Considerations

Serum and plasma are the most common sample types used in rabies serology, but body fluid or purified antibody preparations can be evaluated for presence and level (or potency) of rabies antibodies. Even when the appropriate assay is used, the assigned cut-off may not be applicable to the sample source (species, type, etc.). As an example, an indirect ELISA test using a conjugate that does not equally bind to the Fc portion of the antibody of all species, the cut-off assigned in dogs, may not be applicable for use in cattle samples. Plasma samples containing EDTA are not used in SN methods due to the calcium chelating properties of EDTA, which interferes in the tissue culture cells ability to bind to the slide or well surface. A caveat to always be aware of is the effect of sample quality. Any sample that contains molecules toxic to cells cannot be used in methods using tissue culture cells. Contaminated samples or highly lipemic or hemolyzed samples can interfere in the testing, the effect of each condition is assay dependent. Indirect ELISAs that detect rabies antibody binding to the antigen coated wells via the Fc portion of the antibody, cannot detect F(ab′)2 fragments.

Misunderstanding the capabilities of method can lead to concluding an absence in antibody response if the assay only detects IgG and the sample contains primarily IgM rabies antibodies when collected early in the immune response before immunoglobulin class switching has occurred. In fact, the expectation of ‘a normal range’ of rabies antibody levels arising from rabies vaccination or exposure often does not take into consideration the Ig class, kinetics of response, individual variation in immune genes, and species differences. A study into the levels of the Ig class(es) (IgM and IgG) in relation to the functional RVNA response demonstrated that IgM comprises the bulk of the RVNA response before day 28, the IgG response becomes primary between day 28 and 42, and the kinetics of Ig class peaks are variable per individual [53]. This necessarily impacts choice of cut-off when a study using a method that only detects IgG and includes blood sampling before day 42. This issue is illustrated by the findings of a clinical trial study comparing results of an indirect ELISA and RFFIT performed in parallel that determined equivalent rabies antibody findings were not obtained between assays until day 90 [45]. Rabies vaccine and RIG clinical trials have provided much information on the kinetics of the RVNA response, often reporting the geometric mean titer (GMT) of groups of individuals by time point from initiation of treatment. The GMT provides overall information on the immunogenicity of the vaccine but does not provide the range of responses of the subjects. Figure 2 illustrates the range of individual variation in response to rabies vaccination along with the GMT at different time points, and the difference in kinetics depending on the test method used.

Studies have shown response to vaccination is dependent on an individual’s immune system genes, which can vary considerably, inspiring new vaccine development into targeting antigen epitopes to specific human leukocyte antigen (HLA) genes [54,55,56]. The recognition of high and low responders was noted by Kuwert et al. in 1981, identifying high responders as those with early and high RVNA titer responses compared to lower titers and later peaks in low responders [57]. Low responders tend to have RVNA levels that fall below 0.5 IU/mL within 2–3 years and very low responders to low levels by one year [58,59]. Expecting RVNA levels of 0.5 IU/mL at all blood sample time points, the same antibody peaks and drops for all individuals over time does not reflect reality regardless of what assay is performed. In addition, applying the same expectations of vaccine immunogenicity or response to exposure between species is similarly problematic given the differences in species immune system genes, physiology and in some cases of wildlife, diet (depending on assay type) [21].

## 3. Conclusions

The complexity of rabies antibody measurement is often overlooked as a factor in rabies research, surveillance and vaccine/biologics evaluation. The high fatality rate of rabies, the unique pathogenesis of the virus allowing for vaccination both before and after exposure to effectively protect, as well as laboratory testing capabilities influence interpretation of rabies serology. Problems occur in expectations of antibody levels and their meaning, lack of understanding the performance characteristics of the method used to generate results, as well as the effect of sample type in relation to both. The solution is to review the method validation which defines the capabilities and limitations. Accurate result interpretation requires clearly spelled out purpose of testing which in turn drives selection of the method based on these capabilities and limitations. Equally important is to understand that the assay characteristics are not only unique to the method under consideration, but also must be proven to perform to those characteristic parameters in an individual laboratory. Without method and laboratory qualification, the degree of accuracy, precision, specificity and sensitivity cannot be assumed. The cut-off level selected for a laboratory is dependent on both the laboratory method and the purpose of testing as well as the sample under study. These tenets of clinical laboratory testing are not exclusive in any way to rabies serology. However, verifying immunogenicity of biologics, determination of longevity of protection, and anamnestic responses through RVNA measurement (the closest correlate of rabies immunity currently available) is of critical importance.

## Figures and Tables

**Figure 1 viruses-13-01516-f001:**
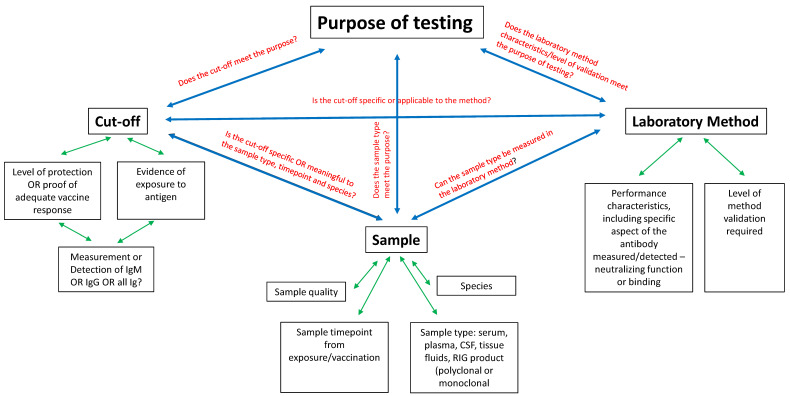
Factors affecting rabies serology interpretation and influences on the main issues of cut-off level, laboratory method, and sample.

**Figure 2 viruses-13-01516-f002:**
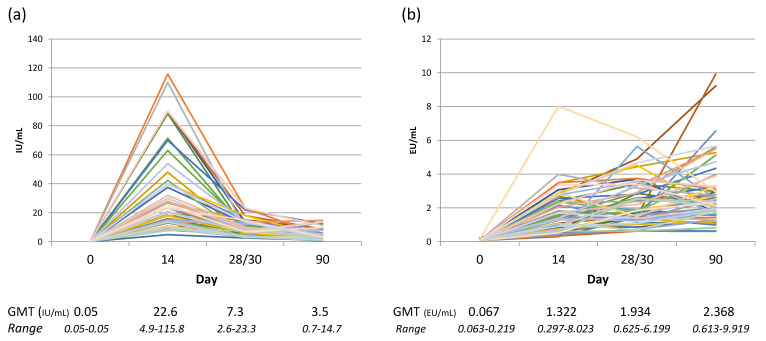
Rabies serology results of human subjects receiving rabies vaccine (PCECV) using the Thai Red Cross regimen, n = 59, the samples were tested by two methods: (**a**) the individual results of RFFIT testing; (**b**) the individual results of indirect ELISA testing. Below the graphs are the GMT and range associated with each time point from the initial vaccine administration. RVNA data from Moore, S. et al. Rabies vaccine response measurement is assay dependent. *Biologicals* **2016**, *44*, 481–486.

## Data Availability

Not Applicable.

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
