# Peer review of "Challenges of Rabies Serology: Defining Context of Interpretation"

_viruses, 2021, doi:10.3390/v13081516_

Round 1

Reviewer 1 Report

Review Report

ID Viruses -1320161

Title: „Challenges of Rabies Serology: Defining Context of Interpretation

Author: Susan M. Moore

Version: 1

Date: 19-07-2021

Reviewer number: 1

A brief summary

The review manuscript „Challenges of Rabies Serology: Defining Context of Interpretation“ by Susan M. Moore describes the issues of interpretation of different rabies serological test results, evaluating as possible criteria for the study of the effectiveness of specific rabies vaccination. As the article is review, its structure is not clearly structured and presented consistently. The purpose of the article is poorly visible in the abstract (L20-22 maybe), but the objective (one - two main sentences) of the Article must be identified at the end of the “Introduction” part (including possible “key-points” – “methods, cut-off and sample selection“). The list of reference should be revised and unified according to the specific recommendation for authors.

Specific comments

L23. Keywords: Rabies Antibody, Serology Interpretation, Assay Comparison

L33-35. Logical clarification is needed: preferably “... Vaccination after exposure and administration of passive immunity in non-vaccinated people, in combination with wound cleaning is highly effective in preventing rabies [7,8].“

L51-52.“...Because RVNA levels are a marker, not the sum of immune protection, ...“ What does this mean? Maybe an expansive explanation is needed?

L61-62. „This level represents seroconversion (not protection),...“ and „RVNA levels are associated with protection, used ... for rabies surveillance...“ (L38)? How to interpret?

L76. „...of these two points...“ maybe “three points” (“method, cut-off and sample selection“), or which  two of this three?

L112-113. „..the level of protection determined in dogs and cats by challenge studies..“ the reference(-s) after “studies” is needed (regardless of the information provided in the following abstract).

L133-134. Very good! Adapted for official regulation (not for scientific-based discussion, Wildlife – not included)…

L150. “ACIP”? The abbreviation explained.

Author Response

Thank you for the review, I believe the comments and suggestions have improved the manuscript.

L23. I added Antibody after Rabies, changed Serology to Serological

L33-35. I have reworded the sentence about  "Vaccination after exposure…..” as suggested.

L51-52. Per the suggestion for expanding on the comment that “RVNA levels are a marker, not sum of protection”, I have added a new sentence and referenced a summary article about the immune response to rabies vaccination and exposure for additional information.

L61-62. Regarding the comment about the dual of use of 0.5 IU/mL as proof of protection and proof of seroconversion, I have added a sentence to clarify that the level was set in human solely based on seroconversion data while it was selected in dogs and cats based on survival data.

L76. Thank you for catching my error in referring to two points when I meant three points, the correction has been made.

L112-113. I added 2 references for the sentence about level of protection in dogs and cats by challenge studies and the level of proof of seroconversion in humans as suggested.

L150. I added the full name for the abbreviation of the first use of ACIP.

Reviewer 2 Report

The manuscript presents current data on serologic testing, highlighting factors important in interpreting results, misdifferentiating results, and properly establishing cut-off points. The manuscript includes a wide range of current knowledge and relevant literature.

In my opinion, the presented aspects of rabies serological rewiev can be published at present form.

Author Response

Thank you for the review and comments.